# Probiotics as Anti-Tumor Agents: Insights from Female Tumor Cell Culture Studies

**DOI:** 10.3390/biom15050657

**Published:** 2025-05-02

**Authors:** Chiara Giorgi, Francesca Lombardi, Francesca Rosaria Augello, Ylli Alicka, Massimiliano Quintiliani, Skender Topi, Annamaria Cimini, Vanessa Castelli, Michele d’Angelo

**Affiliations:** 1Department of Life, Health and Environmental Sciences, University of L’Aquila, 67100 L’Aquila, Italy; chiara.giorgi2@graduate.univaq.it (C.G.); francesca.lombardi@univaq.it (F.L.); massimiliano.quintiliani@gmail.com (M.Q.); annamaria.cimini@univaq.it (A.C.); vanessa.castelli@univaq.it (V.C.); 2Department of Clinical Disciplines, University “Alexander Xhuvani” of Elbasan, 3001 Elbasan, Albania; ylli.alicka@uniel.edu.al (Y.A.); skender.topi@uniel.edu.al (S.T.); 3Sbarro Institute for Cancer Research and Molecular Medicine, Department of Biology, Temple University, Philadelphia, PA 19122, USA

**Keywords:** migration, cancer, cell cycle, triple negative, breast cancer, ovarian cancer

## Abstract

Breast and ovarian cancers are among the most prevalent cancers in women. Cancerous cells are characterized by their ability to continuously cycle and migrate, forming metastases. Some probiotic strains have shown anti-tumorigenic effects. This study tested the impact of probiotics on OVCAR-3 and MDA-MB-231 cell lines by analyzing proteins involved in cell cycle regulation (pP53, Cyclin D1, pERK1), cell survival (AKT), and cell migration (RhoA) using Western blotting and scratch wound tests. Results indicated a reduction in these proteins and decreased cell migration velocity post-treatment. These findings suggest that certain probiotic combinations can arrest the cell cycle, promote cell death, and reduce cell migration, potentially serving as promising candidates alongside standard therapies.

## 1. Introduction

According to the European Cancer Information System, an estimated 374,800 women were diagnosed with breast cancer in 2022, making it the most frequently diagnosed cancer among women in the EU. Breast cancer accounted for nearly 30% of all diagnosed cancers in women, excluding non-melanoma skin cancers. The mortality rate was also significant, with approximately 95,800 deaths, representing 16.7% of all cancer-related deaths in women. According to the American Cancer Society’s 2025 report, breast cancer remains one of the most common cancers among women. The report highlights that the incidence of breast cancer has been rising by about 1% annually from 2012 to 2021. This increase is particularly notable among women under 50 years old, with an annual rise of 1.4%, and among Asian American and Pacific Islander women, with an annual increase of 2.5–2.7%.

Breast cancer, a common cancer affecting women globally, can be categorized into three types: hormone receptor-expressing (estrogen or progesterone receptor), human epidermal receptor 2-positive (HER2+), and triple-negative breast cancer (TNBC), which lacks these receptors.

Ovarian cancer is another prevalent cancer in women, ranking as the fifth leading cause of cancer-related deaths. This high mortality rate is largely due to the fact that ovarian cancer is often diagnosed at an advanced stage, when the disease has already spread beyond the ovaries. Early symptoms are typically vague and can be easily mistaken for less serious conditions, leading to delays in diagnosis and treatment.

Optimizing treatments to improve long-term survival rates is crucial. Current treatment options generally involve a combination of surgery and chemotherapy. Surgery aims to remove as much of the tumor as possible, while chemotherapy targets any remaining cancer cells. Despite these efforts, the five-year survival rate for ovarian cancer remains relatively low, especially for advanced stages of the disease. For instance, the five-year relative survival rate for localized ovarian cancer is about 93%, but it drops to 31% for distant (stage 4) ovarian cancer [1].

A key characteristic of cancer cells is their ability to migrate, allowing them to move within tissues and spread, leading to metastases, the primary cause of cancer-related deaths. Metastases can cause organ failure when cancer cells relocate to secondary sites [2]. Once cancer cells are dislocated to secondary sites, they can cause organ failure [3].

The World Health Organization defines probiotics as live microorganisms that provide health benefits to the host when consumed in adequate amounts [4].

A recent review article discusses how probiotics can help improve the immune system and intestinal health, which are crucial for cancer patients. It highlights that probiotics can enhance the body’s immune response, reduce inflammation, and inhibit the growth of cancer cells by inducing apoptosis in tumor cells.

Specifically, in vitro studies have shown that certain probiotic strains can inhibit the growth of breast cancer cells. For instance, *Lactobacillus acidophilus* and *Bifidobacterium lactis* can induce apoptosis in breast cancer cell lines [5]. Moreover, some studies have shown that the use of *Lactobacillus casei Shirota* could reduce breast cancer incidence in humans [6]. In addition, some probiotics can inhibit the proliferation of cervical cancer cells in vitro [7,8] with *Lactobacillus gasseri* reducing the viability of cervical tumor cells. Probiotics can modulate the immune response and inflammation, which are crucial in the fight against cancer [9,10].

Based on the evidence reported, in this study, we tested the effects of some probiotic strains: *Streptococcus thermophilus*, *Lactobacillus delbrueckii* subsp. *bulgaricus*, *Bifidobacterium lactis*, *Lactobacillus acidophilus*, *Lactobacillus rhamnosus*, *Lactobacillus casei*, and *Lactobacillus plantarum* (see Table 1 for additional information) on two female cancer cell lines: MDA-MB-231 (triple-negative breast cancer) and OVCAR-3 (ovarian adenocarcinoma).

## 2. Materials and Methods

### 2.1. Lysate Preparation

*Streptococcus thermophilus*, *Lactobacillus delbrueckii* subsp. *bulgaricus*, *Bifidobacterium lactis*, *Lactobacillus acidophilus*, *Lactobacillus rhamnosus*, *Lactobacillus casei*, and *Lactobacillus plantarum* in lyophilized form were kindly provided by Dalton Biotecnologie S.r.l. For bacterial sample preparations, each lyophilized probiotic strain was suspended in phosphate-buffered saline (PBS, Euro Clone, West York, UK) at a concentration of 2 × 10^7^ CFU/mL, centrifuged at 8600× *g*, washed twice, and sonicated (30 min, alternating 10 s of sonication and 10 s of pause) using a Vibracell sonicator (Sonic and Materials, Danbury, CT, USA). Bacterial cell disruption was verified by measuring the absorbance of the sample at 590 nm with a spectrophotometer (Eppendorf Hamburg, Germany) before and after every sonication step. The samples, paraprobiotics, were then centrifuged at 18,000× *g*, and the supernatants were filtered with a 0.22 µm pore filter (Corning Incorporated, Corning, NY, USA) to remove any remaining whole bacteria. Total protein content was determined by DC Protein Assay (BioRad, Hercules, CA, USA), using bovine serum albumin (BSA, Sigma Aldrich, St. Louis, MO, USA) as the standard.

### 2.2. Cell Culture

Human cell lines MDA-MB-231 and OVCAR-3 (purchased from ATCC^®^) were cultured with DMEM with 10% fetal bovine serum (FBS, ATCC) and 1% glutamine, without antibiotics, and seeded at 10,000 cells/cm^2^. The fibroblasts, 3T3-L1 (purchased from ATCC), were used as a control and seeded in a 96-multiwell plate and cultured in DMEM (10% FBS, 1% glutamine, without antibiotics). All cells were used as recommended by the manufacturer around passage 10.

### 2.3. BrdU Assay

For the proliferation assay, both cancer cell lines were seeded at 10,000 cell/cm^2^ in a 96-well plate and treated with different concentrations of probiotic lysates (50 µg/mL, 100 µg/mL, and 200 µg/mL proteins) for 24 h in complete medium. Cell proliferation was then evaluated with the Bromodeoxyuridine (BrdU) Cell Proliferation Kit (Abcam, UK), according to the manufacturer’s instructions. BrdU is a thymine analogue that is incorporated in newly synthesized DNA, so it is incorporated in the DNA of proliferating cells, which were then marked using specific anti-BrdU antibodies.

### 2.4. Western Blotting

Western blotting experiments were performed as described before [16]. Briefly, both MDA-MB-231 and OVCAR-3 cells were seeded at a density of 10,000 cells/cm^2^ and treated with the different probiotic strain/strain combinations (paraprobiotics at 100 µg/mL) for 24 h, while the CTR cells of each cell line were treated with the same amount of saline solution. The strain combinations tested were D + E and ALL (all probiotic strains), see Table 1, for 24 h. Samples were lysed with RIPA buffer (Sigma-Aldrich, St. Louis, MO, USA) with freshly added protease and phosphatase inhibitors (Thermo Scientific, Waltham, MA, USA). Protein concentration was evaluated using the BCA Protein Assay Kit (Thermo Scientific), and 30 µg of proteins were run on a 4–12% precast gel and transferred onto a PVDF membrane. The primary antibodies listed below were incubated overnight at +4 °C: anti-pP53 (Cell Signaling 1:1000), anti-p53 (Cell Signaling 1:1000), anti-phospoAKT (Cell Signaling 1:2000), anti-AKT (Cell Signaling 1:1000), anti-phospoERK1/2 (Abcam 1:1000), anti-ERK1/2 (Santa Cruz 1:1000), anti-cyclin D1 (Abcam 1:2000), anti-RhoA (Santa Cruz 1:1000), and anti-β-Actin HRP-conjugated (Cell Signaling 1:1000) antibodies. The incubation of the primary antibodies was followed by 3 TBS-T washes for 10 min each, and the secondary antibodies (Goat-anti Rabbit HRP-conjugate, ThermoFisher Scientific 1:20,000, and Goat-anti Mouse HRP-conjugate, Jackson Immunoresearch research 1:10,000) were incubated for 1 h on oscillation and followed by 4 washes with TBS-T for 10 min each. To detect the chemiluminescence signal, the iBright™ FL1500 Imaging System was used, and the data were processed with Fiji 1.53T software.

### 2.5. Scratch Test

Cell migration was evaluated by performing the Incucyte^®^ Scratch Wound Assay. Briefly, cells were seeded at different densities depending on cell type as suggested by the manufacturer (OVCAR-3 30,000 cells/well, MDA-MB-231 20,000 cells/well, and 3T3-L1 20,000 cell/well in a 96-well plate (BA-04855, Incucyte^®^ Imagelock Plate, Sartorius, Gottingen, Germany)) and allowed to grow to confluence for 48 h. Then, the wound was created using the Incucyte^®^ 96-Well Woundmaker Tool. Cells were washed with sterile PBS 1× (Corning) to remove cell debris and treated with probiotic lysates in complete medium. Plates were then placed into Incucyte^®^ Live-Cell Imaging and Analysis for imaging every 3 h for 24 h. The obtained images were then analyzed with the Incucyte Scratch Wound Analysis Software 2019b rev2.

### 2.6. Cell Viability Assay

Cell viability was evaluated by performing the CellTiter 96^®^ AQueous One Solution Assay. For this assay, 3T3-L1 cells were seeded at 20,000 cells/well in a 96-well plate and allowed to grow to confluence for 24 h. Then, cells were treated with the probiotic combinations (D + E and ALL) and PBS for the control cells for 24 h. After treatments, cells were incubated with 20 µL of CellTiter for 30 min at 37 °C. Finally, the plate was read with the Tecan Spark^®^ spectrophotometer at 492 nm.

### 2.7. Data Analysis

Data were analyzed using GraphPad Prism 8 software (GraphPad, San Diego, CA, USA). All data were presented as the normalized mean ± standard error of the mean (SEM). One-way ANOVA or two-way ANOVA followed by post hoc Tukey–Kramer tests were used for comparison among groups. *p* values < 0.05 were considered statistically significant.

## 3. Results

### 3.1. Anti-Proliferation Activity

The BrdU assay demonstrated that all tested probiotics reduced OVCAR-3 cell proliferation at all concentrations. Strain E was the most effective, reducing proliferation by approximately 60% at 100 µg/mL and 70% at 200 µg/mL. Strains B and C also significantly reduced cell proliferation, with reductions of about 50% at 100 µg/mL and 60% at 200 µg/mL (Figure 1A).

For MDA-MB-231 cells, all probiotics reduced proliferation at all tested concentrations. Strain F was the most effective, causing a 50% reduction in cell proliferation at 200 µg/mL (Figure 1B).

Based on the BrdU assay results, the intermediate concentration of 100 µg/mL was selected for subsequent experiments. Additionally, combinations of D and E probiotic strains (D + E) and all probiotic strains (ALL) were used. Western blot analysis was performed to evaluate the expression of proteins involved in cell proliferation and cell death, including phosphorylated P53 (a tumor suppressor protein), pAKT (an anti-apoptotic protein), pERK1 (involved in cell cycle control), and cyclin D1 (involved in G1 to S phase progression).

The results showed an increase in p-P53 in D + E-treated OVCAR-3 cells and a reduction in ALL-treated OVCAR-3 cells. However, there were no significant changes in MDA-MB-231 cells due to the presence of the mutated form of p53 [17] (Figure 2A,B). Notably, levels of the active form of AKT (phosphorylated form) were significantly reduced by all tested combinations in both cell lines (Figure 2C,D). In parallel, there was a reduction in the levels of p-ERK1 (Figure 2E,F) and cyclin D1 (Figure 2G,H), suggesting that probiotics can activate the tumor suppressor protein and also block the cell cycle at the G1/S phase. While a reduction in the level of phosphorylated (active) ERK1 was observed with probiotic treatments, there was a slight reduction in pERK2 in OVCAR-3 cells and no reduction in pERK2 in the MDA-MB-231 cell line (Figure 2G,H). This could be due to the interdependency of ERK1/ERK2, which can compensate for each other. Moreover, because our experimental conditions showed a reduction in the level of cyclin D1 after both treatments, we can hypothesize that the probiotics in both combinations are effective.

### 3.2. Evaluation of the Effect of Probiotic Lysate Treatment on Cell Migration

To analyze the influence of probiotic treatment on cancer cell migration, a scratch/wound test was performed using the Incucyte^®^ Live-Cell Imaging & Analysis software, evaluating parameters such as width, confluency, and cell density. Both D + E and ALL combinations were able to reduce the migration ability of both tumoral cell lines in terms of the reduction in cell density and confluency; however, the width of the wound did not change with ALL treatment, and it was slightly reduced with D + E treatment (Figure 3A–D), whereas in fibroblasts used as control cells, there was no effect. Moreover, tested strains did not impact cell viability, as reported in Appendix A. Notably, these results suggested an anti-tumorigenic activity of both probiotic combinations tested. Based on these scratch wound-healing experiments, Western blot analysis of RhoA, a protein involved in cell migration, was performed (Figure 4A,B). A reduction in RhoA levels was observed in both cancerous cell lines upon treatment with both probiotic combinations tested, thus confirming the scratch/wound assay results.

## 4. Discussion

Cancer significantly impacts women worldwide, with breast and ovarian cancers being among the most prevalent. Breast cancer is the most commonly diagnosed cancer in women globally, while ovarian cancer ranks as the fifth leading cause of cancer-related deaths [17,18]. The high mortality rate correlated with ovarian cancer is largely due to its diagnosis at advanced stages, as early symptoms are often vague and easily mistaken for less serious conditions. This delay in diagnosis underscores the need for improved screening and early detection methods.

Life expectancy for women with cancer varies significantly based on the type and stage of cancer at diagnosis. For instance, the five-year relative survival rate for localized breast cancer is more than 90% [19,20], but it drops around 40% for distant (stage 4) breast cancer [19]. Similarly, the five-year relative survival rate for localized ovarian stage 1 and 2 cancer is approximately 90–70%, respectively, but it decreases to 20% for distant ovarian cancer (stage 3 and 4) [21]; it is important to consider that only 20% of ovarian cancer is diagnosed in early stages [22]. These statistics highlight the critical importance of early detection and effective treatment strategies to improve long-term survival rates and costs [21,23,24].

Probiotics are live microorganisms that offer health benefits when consumed in sufficient quantities [25,26,27]. Often referred to as “good” or “friendly” bacteria, they help maintain a healthy balance of gut microbiota, which is essential for overall well-being [28,29]. Probiotics can be found in various foods, such as yogurt, sauerkraut, and different types of cheese [30].

Probiotics are renowned for their ability to enhance gut health, aiding in the prevention and treatment of digestive issues like diarrhea, irritable bowel syndrome, and inflammatory bowel disease [29]. They also reinforce the immune system, making it more effective at combating infections and diseases. Recent research indicates that probiotics may positively impact mental health by influencing the gut–brain axis, potentially improving mood and alleviating symptoms of anxiety and depression [31].

In the realm of cancer treatment and prevention, probiotics have shown considerable promise. They can alleviate symptoms associated with cancer treatments, such as chemotherapy-induced diarrhea, by reducing its severity and frequency, thereby enhancing the quality of life for cancer patients [32]. Probiotics can strengthen the body’s immune response, which is vital in fighting cancer, by activating immune cells and dampening inflammation [33]. Certain probiotic strains, like Lactobacillus acidophilus and Bifidobacterium lactis, have been found to inhibit cancer cell growth and induce apoptosis in vitro [34,35]. Maintaining a healthy gut microbiota is crucial for overall health and can influence cancer outcomes, and probiotics help sustain this balance, positively affecting the body’s ability to combat cancer. Furthermore, probiotics can bind to pathogens that convert procarcinogens to carcinogens, thus lowering the risk of cancer development [36]. These advantages highlight the potential of probiotics as a complementary approach in cancer treatment and prevention.

Accordingly, the results of this study provide compelling evidence that specific probiotic strains can significantly inhibit the proliferation and migration of cancer cells, specifically the triple-negative breast cancer cell line MDA-MB-231 and the ovarian adenocarcinoma cell line OVCAR-3. These findings are particularly relevant given the aggressive nature of these cancers and their propensity for metastasis [37].

The BrdU assay results confirmed the anti-proliferative activity of the probiotics, with strains such as E showing a remarkable reduction in cell proliferation. This suggests that certain probiotics can effectively interfere with the cell cycle of cancer cells. An increase in the active form of p53 was observed in OVCAR-3 cells treated with the D + E combination, indicating potential restoration of p53 activity, leading to cell cycle arrest and apoptosis. This is significant as p53 is a crucial tumor suppressor often mutated in cancer cells, resulting in uncontrolled proliferation [38].

Further analysis revealed that the probiotics also impacted other key proteins involved in cell cycle regulation and apoptosis. The reduction in cyclin D1 levels in both cell lines suggested a halt in the cell cycle at the G1/S checkpoint. This is critical, as cyclin D1 is known to be overexpressed in various cancers and is associated with poor prognosis and resistance to treatments, and its level is correlated to tumor size [39,40,41]. The decrease in the active form of AKT supports the hypothesis that probiotics can block apoptosis resistance, enhancing cancer cell death. The reduction in phosphorylated ERK levels aligns with the known role of the ERK1/2 pathway in promoting cell survival and proliferation. By inhibiting this pathway, probiotics may reduce cancer cell survival and proliferation, making them more susceptible to treatment.

The scratch wound assay results demonstrated that D + E and ALL probiotic combinations effectively reduced cell migration in both cancer cell lines. This anti-migration effect was further validated by the reduction in RhoA levels, a protein involved in cell migration and metastasis [42]. These findings suggest that probiotics not only inhibit cancer cell proliferation but also reduce metastatic potential.

Importantly, probiotics did not exhibit harmful effects on non-cancerous cells, indicating a level of specificity towards cancer cells. This specificity is crucial for developing safe and effective cancer therapies.

## 5. Conclusions

This study’s findings suggest that these probiotic strains may counteract the proliferative activity of cancerous cells, highlighting their potential as adjunctive therapies in cancer treatment. The observed reduction in key proteins involved in cell cycle progression and migration underscores the anti-tumorigenic effects of these probiotics.

In conclusion, while the current findings are promising, further research is needed to confirm these results and explore the underlying mechanisms of action. It is essential to acknowledge that our study’s reliance on in vitro models limits the applicability of our findings to clinical settings. In vitro models do not fully capture the complexity of human physiology and tumor microenvironments. Given these limitations, our conclusions should be viewed as preliminary. While the anti-tumor effects of probiotics are promising, they need to be substantiated through further research, particularly in vivo studies that can provide more relevant insights.

Another limitation of this study is that the scope of our research was confined to a select few probiotic strains. This narrow focus may not represent the full spectrum of probiotic effects, necessitating broader investigations to establish more comprehensive conclusions. The necessity to use lysates instead of live probiotics due to sterility procedures in cell culture may represent another limitation.

Future studies should also focus on combining probiotics with standard cancer therapies and investigating their synergistic effects in more complex biological systems to better understand their potential integration into cancer treatment protocols. Moreover, further tests will be necessary to confirm the hypothesis that probiotics can activate apoptosis and induce cell cycle arrest. This includes detailed investigations into the mechanisms by which probiotics influence key regulatory proteins, such as p53 and AKT, which play pivotal roles in apoptosis and cell cycle control.

## Figures and Tables

**Figure 1 biomolecules-15-00657-f001:**
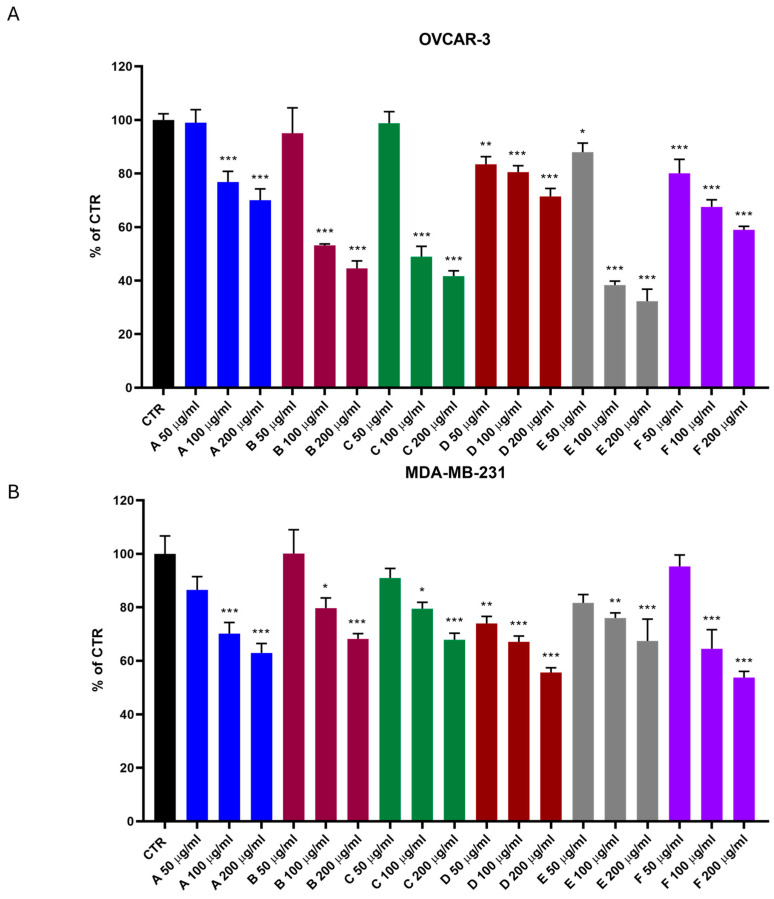
(**A**) BrDU proliferation assay on human ovarian adenocarcinoma OVCAR-3. One-way ANOVA ***, *p* < 0.0005; **, *p* < 0.005; *, *p* < 0.05 vs. CTR. (**B**) BrDU proliferation assay on human triple-negative breast cancer MDA-MB-231. One-way ANOVA ***, *p* < 0.0005; **, *p* < 0.005; *, *p* < 0.05 vs. CTR.

**Figure 2 biomolecules-15-00657-f002:**
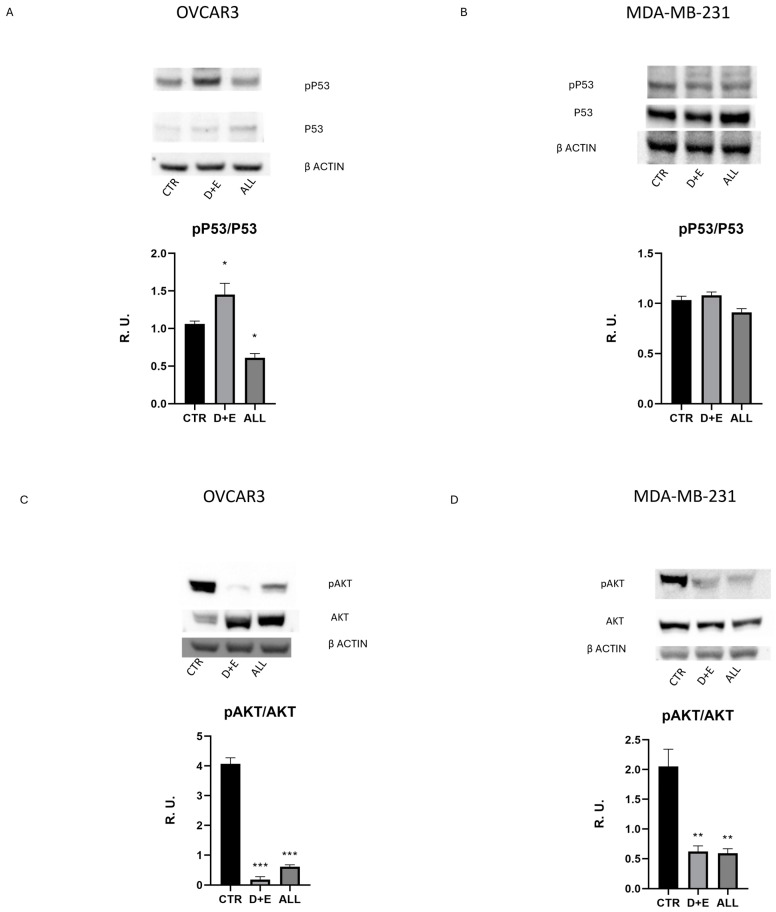
Western blotting of phosphorylated P53(original images can be found in supplementary) (**A**,**B**), phosphorylated AKT (**C**,**D**), phosphorylated ERK1/2 (**E**,**F**), and cyclin D1 (**G**,**H**) in both cancer cell lines. The data are expressed as mean ± SEM. For the comparative data analysis, ANOVA followed by Dunnett’s test was performed. * *p* < 0.05; ** *p* < 0.005; *** *p* < 0.0005.

**Figure 3 biomolecules-15-00657-f003:**
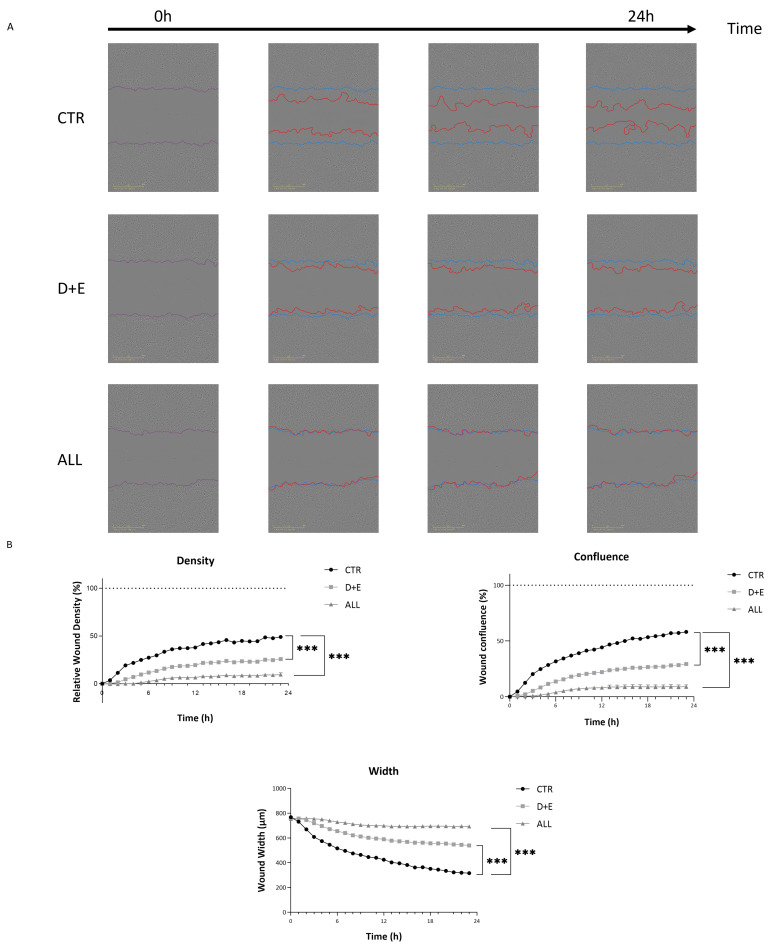
Scratch wound test of OVCAR-3 (**A**) and MDA-MB-231 (**C**) frames at different times (from 0 h to 24 h). Scratch wound test of OVCAR-3 (**B**) and MDA-MB-231 confluency, density, and amplitude graphs (**D**). Data are expressed as mean ± SEM. For the comparative data analysis, a two-way ANOVA followed by Dunnett’s test was performed. *** *p* < 0.0005.

**Figure 4 biomolecules-15-00657-f004:**
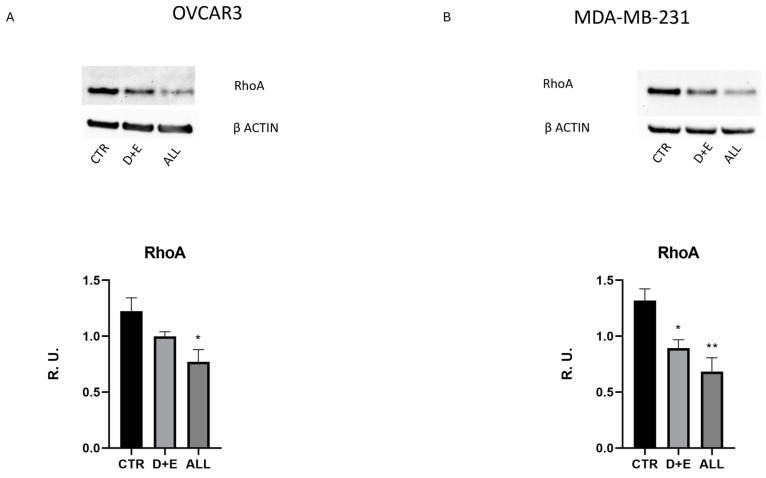
Western blotting of RhoA (original images can be found in supplementary) (**A**,**B**) in both cancer cell lines. The data are expressed as mean ± SEM. For the comparative data analysis, ANOVA followed by Dunnett’s test was performed. * *p* < 0.05; ** *p* < 0.005.

**Table 1 biomolecules-15-00657-t001:** Information about the tested probiotic strains.

ID	Species	Characteristics
A	*Streptococcus thermophilus*	It is safely used in food; it has the Qualified Presumption of Safety status in the EU. Among the health benefits it has, we can find antioxidant compound production, anti-inflammatory effects, risk alleviation for some cancers, and antimutagenic effects [11].
B	*Lactobacillus delbrueckii* subsp. *bulgaricus*	It has been demonstrated to be able to inhibit the development of colitis-associated cancer in mice, attenuating intestinal inflammation [12].
C	*Bifidobacterium lactis*	It is in commercialized yogurt, and it has been demonstrated that it has a beneficial role in maintaining intestinal barrier function and that it supports normal physiological function in immunosenescent elderly [13].
D	*Lactobacillus acidophilus*	Often used in yogurts, it has antimutagenic properties, anticarcinogenic properties [14], anti-diarrheal properties, and immune system stimulation properties, and many other beneficial effects, such as maintenance of balanced flora and improvement in lactose metabolism [15].
E	*Lactobacillus rhamnosus*	Used as a treatment in animal models, it may reduce the risk of colon cancer by modulating the gut microbiota and downregulating pro-inflammatory molecules [4].
F	*Lactobacillus casei*	Often used in yogurts, it has antimutagenic properties, anticarcinogenic properties [14], anti-diarrheal properties, and immune system stimulation properties, and many other beneficial effects, such as maintenance of balanced flora and improvement in lactose metabolism [15].

## Data Availability

Data will be provided upon reasonable requests by contacting the corresponding author.

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
