# Peer review of "Probiotics as Anti-Tumor Agents: Insights from Female Tumor Cell Culture Studies"

_biomolecules, 2025, doi:10.3390/biom15050657_

Round 1
Reviewer 1 Report
Comments and Suggestions for Authors
The manuscript presents intriguing findings on probiotic-mediated anti-cancer effects in human breast and ovarian carcinoma models using proliferation assay, migration test, and Western blotting. However, complementary studies are required to confirm that single and combinatorial treatments with probiotics induce anti-cancer effects in vitro. I strongly recommend that the authors perform the following:
- Viability assays must be conducted on normal 3T3-L1 cells to document potential side effects, and scratch assay alone is insufficient to draw such conclusions.
- The claim that probiotics reduce cell viability needs to be confirmed by Annexin V-FITC and PI staining. Additionally, reduced cell migration requires confirmation through invasion assay.
- Since the aim of the study was to determine the apoptosis-inducing and anti-metastatic effects of probiotics, the expression of more relevant proteins, such as BCL-2, BAX, and MMPs, is recommended.
- Performing Western blotting on RhoA expression does not necessarily indicate its reduced activity. More functional analyses, such as gelatin and casein zymography, are recommended to determine anti-metastatic effects via MMP activity, rather than protein expression.
- It would be beneficial if the authors extended the treatment time to include 48 and 72 hours to provide a more comprehensive understanding of the effects over time.
Author Response
Reviewer 1:
The manuscript presents intriguing findings on probiotic-mediated anti-cancer effects in human breast and ovarian carcinoma models using proliferation assay, migration test, and Western blotting. Response: We really appreciate the Reviewer’s comments, and the time spent in revising our review article. We tried to do our best to address all the points raised.
However, complementary studies are required to confirm that single and combinatorial treatments with probiotics induce anti-cancer effects in vitro. I strongly recommend that the authors perform the following:
- Viability assays must be conducted on normal 3T3-L1 cells to document potential side effects, and scratch assay alone is insufficient to draw such conclusions.
- The claim that probiotics reduce cell viability needs to be confirmed by Annexin V-FITC and PI staining. Additionally, reduced cell migration requires confirmation through invasion assay.
- Since the aim of the study was to determine the apoptosis-inducing and anti-metastatic effects of probiotics, the expression of more relevant proteins, such as BCL-2, BAX, and MMPs, is recommended.
- Performing Western blotting on RhoA expression does not necessarily indicate its reduced activity. More functional analyses, such as gelatin and casein zymography, are recommended to determine anti-metastatic effects via MMP activity, rather than protein expression.
- It would be beneficial if the authors extended the treatment time to include 48 and 72 hours to provide a more comprehensive understanding of the effects over time.
Response: We appreciate the reviewer's comments and agree that further investigations are needed. However, we did our best to adhere to the project schedule with the grant amount received. Consequently, we were able to perform the viability assay for 3T3-L1 as suggested. Unfortunately, we are currently unable to conduct the other requested assays due to time constraints and the lack of necessary materials in our laboratory. We hope the reviewer understands our situation.
Reviewer 2 Report
Comments and Suggestions for Authors
The study with the title Probiotics as anti-tumor agents: insights from female tumor cell culture studies, evaluated the effects of probiotics on OVCAR-3 (ovarian cancer) and MDA-MB-231 (breast cancer) cell lines by assessing key proteins involved in cell cycle regulation (pP53, Cyclin D1, pERK1), cell survival (AKT), and migration (RhoA) through western blotting and scratch wound assays.
Probiotic treatment reduced these proteins' expression and decreased cell migration, indicating cell cycle arrest and impaired migratory capacity. These findings highlight the potential of specific probiotics to suppress cancer cell proliferation and mobility, supporting their role as adjuncts to conventional cancer therapies.
Further research should explore the combined effects of probiotics and chemotherapeutic agents, such as paclitaxel, to determine potential synergistic benefits. Additionally, mechanistic studies are needed to confirm the ability of probiotics to induce apoptosis and enforce cell cycle arrest by modulating critical regulatory proteins like p53 and AKT. Such evidence could strengthen the therapeutic value of probiotics in integrative cancer treatment strategies.
Author Response
Reviewer 2:
The study with the title Probiotics as anti-tumor agents: insights from female tumor cell culture studies, evaluated the effects of probiotics on OVCAR-3 (ovarian cancer) and MDA-MB-231 (breast cancer) cell lines by assessing key proteins involved in cell cycle regulation (pP53, Cyclin D1, pERK1), cell survival (AKT), and migration (RhoA) through western blotting and scratch wound assays.
Probiotic treatment reduced these proteins' expression and decreased cell migration, indicating cell cycle arrest and impaired migratory capacity. These findings highlight the potential of specific probiotics to suppress cancer cell proliferation and mobility, supporting their role as adjuncts to conventional cancer therapies.
Further research should explore the combined effects of probiotics and chemotherapeutic agents, such as paclitaxel, to determine potential synergistic benefits. Additionally, mechanistic studies are needed to confirm the ability of probiotics to induce apoptosis and enforce cell cycle arrest by modulating critical regulatory proteins like p53 and AKT. Such evidence could strengthen the therapeutic value of probiotics in integrative cancer treatment strategies.
Response: We really appreciate the Reviewer’s comments and the time spent in revising our review article. We totally agree with the reviewer and stressed that further investigations are needed, including other mechanistic studies.
Reviewer 3 Report
Comments and Suggestions for Authors
The article entitled “Probiotics as anti-tumor agents: insights from female tumor cell 2 culture studies” is well-written and easy to understand.
The main question raised in the article concerns the effect of probiotics as anti-tumor agents on female tumor cell culture.
The topic is not original but very important because anything that can help fight cancer, slow it down, and limit it is worth attention.
In my opinion, this paper is suitable for publication in the Biomolecules but needs to be improved.
- Page 3, line 111: What are the CTR cells?
- Page 3, line 112: Why were such combinations of strains chosen for this study, and why were only two combinations studied? In this sentence it is written "D+E" and in brackets "D, E, G" and then in the text only "D+E" is mentioned, there is no mention of strain G. Which information is correct? Please standardize this record, both in the text and in the figures. Please place "Table 1" after the record "(all probiotic strains)" and before "for 24 hours", to clarify the origin of the strain designations.
- Please, in all drawings, correct the captions indicating control, because sometimes it is written CTR and other times CTRL.
- The Discussion section needs more references to support the content.
Author Response
Reviewer 3:
The article entitled “Probiotics as anti-tumor agents: insights from female tumor cell 2 culture studies” is well-written and easy to understand.
The main question raised in the article concerns the effect of probiotics as anti-tumor agents on female tumor cell culture.
The topic is not original but very important because anything that can help fight cancer, slow it down, and limit it is worth attention.
In my opinion, this paper is suitable for publication in the Biomolecules but needs to be improved.
Response: We really appreciate the Reviewer’s comments, and the time spent in revising our review article. We tried to do our best to address all the points raised.
- Page 3, line 111: What are the CTR cells?
Response: We apologize for the oversight, an explanation on what the CTR cells are has been added.
- Page 3, line 112: Why were such combinations of strains chosen for this study, and why were only two combinations studied? In this sentence it is written "D+E" and in brackets "D, E, G" and then in the text only "D+E" is mentioned, there is no mention of strain G. Which information is correct? Please standardize this record, both in the text and in the figures. Please place "Table 1" after the record "(all probiotic strains)" and before "for 24 hours", to clarify the origin of the strain designations.
Response: we thank the reviewer for the comments. Correction on the kind of probiotics used has been made and the “Table 1” has been added to the text as suggested by Reviewer 3.
- Please, in all drawings, correct the captions indicating control, because sometimes it is written CTR and other times CTRL.
Response: All the “CTRL” have been corrected into “CTR” as suggested.
- The Discussion section needs more references to support the content.
Response: We appreciate the Reviewer’s comment and we added more references to support the content as suggested.
Reviewer 4 Report
Comments and Suggestions for Authors
Dear Authors, the idea of your paper is interesting and worth of study... but...
some issues need to be imrpoved or precised
- Inactive bacteria, used in lysate fail probiotic definition. It's called paraprobiotic.
- did you use a fresh cell line or passaged?
- did you use any antibiotic in cell assays?
- did you consider to use any negative control like no-cancer human cell line (i.e. fibroblast) to check the effect of high concentration of lysates on cells?
- The Discussion is generally based on your own observations without references to other studies.
The language could be improved
Author Response
Reviewer 4:
Dear Authors, the idea of your paper is interesting and worth of study... but...some issues need to be improved or precised
Response: We really appreciate the Reviewer’s comments, and the time spent in revising our review article. We tried to do our best to address all the points raised.
- Inactive bacteria, used in lysate fail probiotic definition. It's called paraprobiotic.
Response: we appreciate the reviewer’s comment and we agree. The “paraprobiotics” word as been added in the definition as suggested by reviewer 4.
- did you use a fresh cell line or passaged?
Response: This information has been added as requested by reviewer 4.
- did you use any antibiotic in cell assays?
Response: This information has been added as requested by reviewer 4.
- did you consider to use any negative control like no-cancer human cell line (i.e. fibroblast) to check the effect of high concentration of lysates on cells?
Response: The no-cancer cell line used was the 3T3-L1 fibroblast and a cell viability assay other than migration assay has been added as supplementary figure.
- The Discussion is generally based on your own observations without references to other studies.
Response: We thank the Reviewer for the comment. More references to other studies have been added to the discussion section as suggested.
Round 2
Reviewer 4 Report
Comments and Suggestions for Authors
thank you for the improvements.
Author Response
We really appreciate the comments of the reviewer that greatly improved our research article.